# Uncovering Overfitting in Large Language Model Editing

**Mengqi Zhang[1]\*, Xiaotian Ye[2]\*, Qiang Liu[3], Shu Wu[3]†, Pengjie Ren[1]†, Zhumin Chen[1]**
[1]Shandong University
[2]School of Computer Science, Beijing University of Posts and Telecommunications
[3]New Laboratory of Pattern Recognition (NLPR)
State Key Laboratory of Multimodal Artificial Intelligence Systems (MAIS)
Institute of Automation, Chinese Academy of Sciences
{mengqi.zhang, renpengjie, chenzhumin}@sdu.edu.cn
yexiaotian@bupt.edu.cn,{qiang.liu, shu.wu}@nlpr.ia.ac.cn

## Abstract

Knowledge editing has been proposed as an effective method for updating and correcting the internal knowledge of Large Language Models (LLMs). However, existing editing methods often struggle with complex tasks, such as multi-hop reasoning. In this paper, we identify and investigate the phenomenon of **Editing Overfit**, where edited models assign disproportionately high probabilities to the edit target, hindering the generalization of new knowledge in complex scenarios. We attribute this issue to the current editing paradigm, which places excessive emphasis on the direct correspondence between the input prompt and the edit target for each edit sample. To further explore this issue, we introduce a new benchmark, EVOKE (EValuation of Editing Overfit in Knowledge Editing), along with fine-grained evaluation metrics. Through comprehensive experiments and analysis, we demonstrate that Editing Overfit is prevalent in current editing methods and that common overfitting mitigation strategies are ineffective in knowledge editing. To overcome this, inspired by LLMs' knowledge recall mechanisms, we propose a new plug-and-play strategy called Learn the Inference (LTI), which introduce a Multi-stage Inference Constraint module to guide the edited models in recalling new knowledge similarly to how unedited LLMs leverage knowledge through in-context learning. Extensive experimental results across a wide range of tasks validate the effectiveness of LTI in mitigating Editing Overfit.

## 1 Introduction

Large Language Models (LLMs) have achieved remarkable success across various Natural Language Processing (NLP) tasks (Zhao et al., 2023), yet they often contain outdated or incorrect information, raising concerns about their reliability and factual accuracy. Knowledge Editing (Yao et al., 2023) has emerged as a promising solution to precisely update or correct a model's knowledge. Among the different editing strategies, parameter-modifying methods, which directly alter the model's internal parameters, have garnered significant attention from the research community. These include fine-tuning-based techniques such as FT-L (Zhu et al., 2020), meta-learning approaches like KE (De Cao et al., 2021) and MEND (Mitchell et al., 2021), and locate-then-edit techniques such as ROME (Meng et al., 2022a) and MEMIT (Meng et al., 2022b).

Although existing methods have achieved promising results, their performance experiences a catastrophic decline when transferred to complex tasks involving reasoning (Yao et al., 2023). For instance, in the representative multi-hop reasoning task, after the LLM is updated with *Steve Jobs* as *the founder of Microsoft*, it can easily respond to straightforward questions like "*Who is the founder of Microsoft?*" with "*Steve Jobs.*" However, it struggles to accurately answer more complex queries, such as "*Which college did the founder of Microsoft attend?*"

---

*\*Equal contribution.*
*†Corresponding authors.*

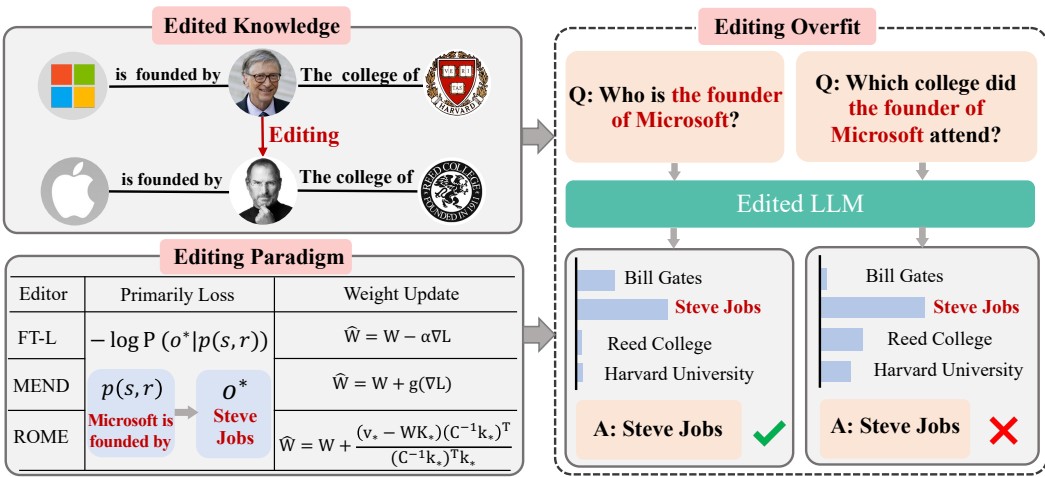

Figure 1: Example of Editing Overfit.

To investigate the reasons behind the failure of edited LLMs in complex tasks, we first experimentally analyse the outputs from edited models on a multi-hop reasoning task (§3). The results reveal an abnormally high probability that the edited models output the edit target $o^*$ for multi-hop questions, even when such responses are entirely implausible as valid answers (§3.2). We refer to this phenomenon as **Editing Overfit, indicates that edited models tend to assign unusually high prediction probabilities to the edit target $o^*$ of edit sample $(s, r, o, o^*)$, skewing the response accuracy for complex questions where the correct answer is not $o^*$.** For instance, as shown in Figure 1, after editing "*Microsoft is founded by Bill Gates → Steve Jobs,*" it erroneously answers the question "*Which college did the founder of Microsoft attend?*" with "*Steve Jobs.*"

We hypothesize that Editing Overfit is a key factor contributing to the suboptimal performance of edited LLMs on complex tasks, like multi-hop editing. **This phenomenon likely stems from existing knowledge editing paradigms emphasize the direct correspondence between the input prompt $p(s, r)$ and the output $o^*$ for each edit sample $(s, r, o, o^*)$. Given the typically limited number of optimization samples, this focus on optimizing the $p(s, r) \rightarrow o^*$ relationship can lead to severe overfitting issues.** Specifically, as shown in Figure 1, all current editing methods for LLMs rely on a primary loss function that maximizes the likelihood of the new target $o^*$ given the input prompt $p(s, r)$. The main differences between these methods lie in the techniques used for parameter updates. For example, FT-based methods either directly optimizes or uses parameter-efficient fine-tuning (Hu et al., 2022; Ren et al., 2024) to adjust model parameters, MEND employ a hypernetwork to make updates, while ROME and MEMIT apply low-rank updates to derive closed-form solutions for specific parameters. When the model is updated with the new knowledge such as "*Microsoft is founded by Steve Jobs,*" it risks overfitting by learning only the correspondence between "*Microsoft is founded by*" and "*Steve Jobs.*" As a result, the edited model may output "*Steve Jobs*" whenever it encounters the terms "*Microsoft*" and "*is founded by.*" This also explains the abnormally high prediction probabilities of edit targets in multi-hop reasoning task, as the edited model may simply recognize patterns in the prompt and tend to output the corresponding edit target.

In this study, we particularly investigate the Editing Overfit phenomenon that occurs in edited LLMs. To this end, we first construct a benchmark for **EV**aluating of Editing **O**verfit in **K**nowledge **E**diting (EVOKE) (§4.1), which comprises six tasks across two categories. The overfit tasks in EVOKE include various patterns prone to causing overfitting in models, allowing us analyze and investigate overfitting phenomena in current editing methods. By applying existing editing methods to EVOKE, we conduct an in-depth analysis to identify specific input patterns are prone to overfitting (§4.2). Furthermore, we evaluate the effectiveness of four existing overfitting mitigation strategies (§5), *Norm Constraints*, *Batch Editing*, *Multi-layer Editing*, and *Data Augmentation*, in addressing the Editing Overfit problem.

To further alleviate Editing Overfit, inspired by the knowledge mechanism of LLMs, we propose a plug-and-play strategy named **L**earn **T**o **I**nference (LTI) (§6), which enables the edited models to learn how to infer with new knowledge rather than simply establish input-output mappings. Specifically, LTI introduces a Multi-Stage Constraint module, which imposes constraints on crucial

reasoning steps of LLMs during the editing process. This ensures that the edited model utilizes new knowledge in a way that closely resembles how an unedited model leverage new knowledge through in-context learning, helping to prevent the model from overfitting solely on input-output mapping. Additionally, LTI can be combined with various knowledge editing methods and used in conjunction with other overfitting mitigation techniques.

Our contributions can be summarized as follows:

- We reveal and investigate the overfitting issue caused by current editing paradigm, identifying it as a key factor behind the suboptimal performance of edited models, a phenomenon we term the Editing Overfit problem.
- We construct EVOKE, a benchmark with detailed evaluation metrics, to enable a fine-grained assessment and analysis of mainstream editing methods. Additionally, we explore the effectiveness of four general overfitting mitigation techniques in addressing the Editing Overfit problem.
- We propose a new plug-in strategy, Learn the Inference, designed to further mitigate overfitting. Extensive experiments demonstrate that integrating LTI with different editing methods effectively reduces the severity of Editing Overfit.

## 2 RELATED WORK

Knowledge editing (KE) updates LLM outputs to (i) accurately respond to new knowledge, (ii) preserve existing knowledge without catastrophic forgetting, and (iii) leverage updated knowledge in complex reasoning tasks. Each piece of knowledge is formulated as a triple $(s, r, o)$ (De Cao et al., 2021), consisting of a subject $s$, relation $r$, and object $o$. An edit sample is defined as $e = (s, r, o, o^*)$, representing a knowledge update from $(s, r, o)$ to $(s, r, o^*)$. Our study focuses on parameter-modifying methods, which are divided into three main categories (Yao et al., 2023):

**Fine-tuning-based methods** generally follow the supervised fine-tuning paradigm. For example, to edit a fact such as *"Microsoft is founded by Steve Jobs,"* the model's weights are updated via gradient descent to increase the probability of the edit target, *Steve Jobs*. Some approaches aim to improve robustness by incorporating norm constraints (Zhu et al., 2020) or data augmentation(Gangadhar & Stratos, 2024; Wei et al., 2024). However, vanilla fine-tuning often affects unrelated knowledge, leading to catastrophic forgetting, making it unsuitable for direct application in knowledge editing.

**Meta-learning-based methods** employ a hypernetwork to adjust model parameters specifically for editing. This hypernetwork is trained to convert fine-tuning gradients into updated weights, with the aim of predicting weights that closely resemble those obtained through fine-tuning with augmented data. KE (De Cao et al., 2021) pioneered this approach, which MEND (Mitchell et al., 2021) later extended to LLMs by predicting low-rank decompositions of parameter updates.

**Locate-then-edit methods** originate from research into the internal mechanisms of LLMs, advocating for identifying the specific weights responsible for storing knowledge before applying targeted updates. Geva et al. (2021; 2023) propose viewing MLP modules as key-value memory. Building on this foundation, the Knowledge Neuron theory (Dai et al., 2022) posits that these MLP key-value pairs encode factual knowledge. Meng et al. (2022a) introduce causal tracing to analyze LLMs' factual recall mechanisms, leading to the development of ROME (Meng et al., 2022a) and MEMIT (Meng et al., 2022b), which achieved state-of-the-art results on several traditional metrics.

In recent years, researchers have recognized the limitations of current editing methods on specific complex tasks such as multi-hop reasoning, leading to the development of task-specific approaches (Zhong et al., 2023; Zhang et al., 2024b;a). More detailed related work is provided in Appendix B. In contrast, our work explores the reasons behind the suboptimal performance of editing methods by constructing a benchmark and proposes a more general strategy to enhance editing performance by addressing the issue of overfitting.

## 3 PRELIMINARY EXPERIMENTS

To investigate the causes of edited LLMs' poor performance on complex tasks, we begin by analyzing the outputs of the edited models on a representative multi-hop reasoning dataset, COUNTER-

FACTPLUS (Yao et al., 2023), where each entry contains an edited knowledge $e = (s, r, o, o^*)$ along with a multi-hop question $q = (s, r, r')$ that requires reasoning based on the edited sample.

## 3.1 METRIC DEFINITIONS

To perform a fine-grained analysis of the outputs from edited models, we define several metrics in response to complex prompts, such as multi-hop questions within the dataset. Specifically, for each edit sample $e = (s, r, o, o^*)$, when the edited LLM is presented with a prompt consisting of a complex question, it may produce one of the following outputs: the original answer to the complex question, the correct answer, or the edited target $o^*$. Accordingly, we define the following metrics:

- **Correct Answer Probability (CAP)**: The probability that the model generates the correct answer `ans` for a given `prompt`, formalized as $\mathbb{P}(\texttt{ans} \mid \texttt{prompt})$.

- **Original Answer Probability (OAP)**: The probability that the model outputs the original answer `ori` (before editing) in response to the given `prompt`, defined as $\mathbb{P}(\texttt{ori} \mid \texttt{prompt})$.

- **Direct Probability (DP)**: The likelihood that the model produces the edit target $o^*$, expressed as $\mathbb{P}(o^* \mid \texttt{prompt})$.

To further evaluate the influence of both the target edit $o^*$ and the original answer `ori` on the correct answer `ans`, we follow Meng et al. (2022a) and define two additional comprehensive metrics to gauge the model's overall editing effectiveness:

- **Editing Overfit Score (EOS)**: This metric evaluates the performance of the edited model on complex questions where the correct answer is not $o^*$. It serves as a primary indicator of the model's overfitting and overall performance. The score is calculated as the proportion of cases where the model overfits by favoring the edit target $o^*$ over the correct answer `ans`, formalized as $\mathbb{E}\left[\mathbb{I}[\mathbb{P}(\texttt{ans} \mid \texttt{prompt}) > \mathbb{P}(o* \mid \texttt{prompt})]\right]$.

- **Answer Modify Score (AMS)**: This metric evaluates the negative interference of old knowledge on the correct answers. It is assessed by calculating the proportion of cases where the probability of the correct answer exceeds that of the original answer, defined as $\mathbb{E}\left[\mathbb{I}[\mathbb{P}(\texttt{ans} \mid \texttt{prompt}) > \mathbb{P}(\texttt{ori} \mid \texttt{prompt})]\right]$.

## 3.2 EDITING OVERFIT PHENOMENON

Subsequently, we apply the ROME and MEMIT methods to GPT-J to evaluate the performance of the edited models on COUNTER-FACTPLUS using the aforementioned metrics, as shown in Figure 2. In multi-hop evaluations, the edit target $o^*$ for each edit sample $(s, r, o, o^*)$ is typically not a possible answer to the multi-hop prompt, and its output probability should therefore be negligible. For instance, "*Steve Jobs*" would be an implausible response to "*Which college did the founder of Microsoft*

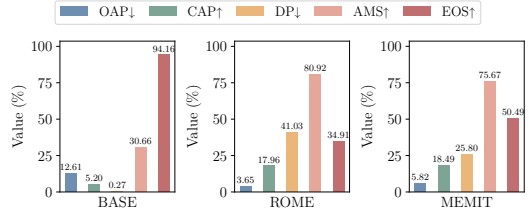

Figure 2: Performance of GPT-J edited with ROME and MEMIT on COUNTERFACTPLUS.

*attend?*" The base model's DP score of $0.27\%$ confirms that the unedited model is highly unlikely to output $o^*$ as a response. However, after editing, both models exhibit significantly higher average probabilities of $o^*$ (DP), with ROME even reaching $41.03\%$. Both models also show substantially lower Editing Overfit Score (EOS) values, indicating that for many evaluation samples, the probability of generating the correct answer is lower than that of outputting $o^*$. This anomalous probability distribution substantially impacts model performance, as the inflated $o^*$ prediction probability diminishes the Correct Answer Probability (CAP) and obscures the model's actual output.

From these observations, we define the phenomenon of **Editing Overfit** as follows: **After an LLM has been edited based on an editing example $e = (s, r, o, o^*)$, the edited LLM exhibits a heightened likelihood of producing the edit target $o^*$ as the answer to questions that implicitly or explicitly contains $s$ or $r$, even when the correct answer is unrelated to $o^*$.**

## 4 ANALYSIS ON EDITING OVERFIT

To further investigate the severity of Editing Overfit in edited LLMs, we construct EVOKE, a new benchmark designed to analyze overfitting phenomena across various tasks. We then assess the performance of different editing methods using this benchmark and examine the effectiveness of several existing mitigation strategies in reducing Editing Overfit.

### 4.1 EVOKE BENCHMARK

EVOKE comprises Recall Tasks and Overfit Tasks, covering six tasks in total. The Recall Tasks assess the edited model's ability to recall new edited knowledge, including **Efficacy** and **Paraphrase** evaluation. The Overfit Tasks pose complex challenges that are prone to inducing overfitting in editing methods, including **Multi-hop Reasoning**, **Prefix Distraction**, **Subject Specificity**, and **Relation Specificity**. These tasks are specifically designed to evaluate the model's capability to utilize newly integrated knowledge for more challenging scenarios, with a particular emphasis on examining the degree of Editing Overfit. Details of EVOKE construction can be found in Appendix C.

Taking the edit "*Microsoft is founded by Bill Gates → Steve Jobs*" as an example, we introduce the recall tasks used to assess editing success rate of the edit (Meng et al., 2022a; Yao et al., 2023):

- **Efficacy** directly validates whether the edited models can recall the new edited knowledge $(s, r, o^*)$ under the editing prompt $p(s, r)$. In the context of the above example, the model would be asked: "*Who is the founder of Microsoft?*"

- **Paraphrase** examines the model's ability of recall the new knowledge $(s, r, o^*)$ using paraphrased forms of the editing prompt $p(s, r)$. For instance, it might ask:" *Who established Microsoft?*"

The design of overfit tasks are based on the two principles: First, the input questions explicitly or implicitly contain the information of subject $s$ or relation $r$ to induce potential overfitting responses from the model; Second, the correct answers to these questions are entirely unrelated to $o*$, making it easier to determine whether the edited model exhibits overfitting. Accordingly, the overfit tasks are constructed as follows:

- **Multi-hop Reasoning** evaluates the edited model's ability to integrate the newly edited knowledge with existing knowledge to correctly answer questions spanning multiple entities or relations. For example, "*Which university did the founder of Microsoft attend?*" These questions typically contain implicit subject $s$ and relation $r$ information from the edit sample, but the answer is not the target $o^*$. They are well-suited for evaluating whether the edited model has overfit to the $p(s, r) \rightarrow o^*$ pattern. A model that has overfit to this pattern might incorrectly produce '*Steve Jobs*' as the answer to this question.

- **Prefix Distraction** uses the new knowledge $(s, r, o^*)$ as a perfix for unrelated questions, evaluating weather the edited model can still provide the original correct answer. For example: "*Microsoft was founded by Steve Jobs. Who is the founder of Amazon?*" This evaluation also assess weather the edited model has overfit to the $p(s, r) \rightarrow o^*$ pattern, providing a more explicit measure compared to multi-hop reasoning.

- **Subject Specificity** presents questions with the same subject $s$ as the edit sample but with different relations $r'$. For example: "*When was Microsoft founded?*" These questions typically contain information about the subject $s$, but the correct answer is not the target $o^*$, making them ideal for evaluating whether the edited model has overfit to the $s \rightarrow o^*$ pattern.

- **Relation Specificity** includes questions with different subjects $s'$ from the edit sample but the same relation $r$, such as: "*Who is the founder of Amazon?*" These questions contain information about the relation $r$, but the answer is not the target $o^*$. They are used to evaluate whether the model has overfit to the $r \rightarrow o^*$ pattern. This task also corresponds to the locality evaluation in COUNTERFACT (Meng et al., 2022a).

The recall task is evaluated using the AMS metric. For the multi-hop reasoning task, we employ all five metrics defined in Section 3.1 for a comprehensive analysis. In the Prefix Distraction, Subject Specificity, and Relation Specificity tasks, the correct answer is identical to the original answer, making OAP equivalent to CAP, with the EOS metric used to evaluate performance in these tasks.

Table 1: Experimental results for different models on the Overfit Tasks of EVOKE.

| Editor | Prefix Distraction | | | Multi-hop Reasoning | | | | | Subject Specificity | | | Relation Specificity | | |
|---|---|---|---|---|---|---|---|---|---|---|---|---|---|---|
| | DP↓ | CAP↑ | EOS↑ | DP↓ | OAP↓ | CAP↑ | EOS↑ | AMS↑ | DP↓ | CAP↑ | EOS↑ | DP↓ | CAP↑ | EOS↑ |
| GPT-2 XL | 5.01 | 13.30 | 54.58 | 0.27 | 7.87 | 4.21 | 93.43 | 35.64 | 0.50 | 4.90 | 85.59 | 0.33 | 6.42 | 79.61 |
| FT | 22.97 | 9.05 | 23.40 | 4.57 | 4.10 | 7.06 | 75.91 | 66.91 | 2.80 | 4.20 | 51.31 | 11.99 | 5.07 | 40.81 |
| FT-L | 12.87 | 11.00 | 40.09 | 1.60 | 7.13 | 6.19 | 87.71 | 50.97 | 0.63 | 4.47 | 77.07 | 2.91 | 6.31 | 70.12 |
| MEND | 46.57 | 1.93 | 17.88 | 3.27 | 4.75 | 7.42 | 83.82 | 40.15 | 10.37 | 4.62 | 36.03 | 11.99 | 5.43 | 44.65 |
| ROME | 44.99 | 6.62 | 15.22 | 23.32 | 3.43 | 11.76 | 46.11 | 75.67 | 35.39 | 2.60 | 21.83 | 1.01 | 6.47 | 77.23 |
| ROME-LTI | 19.53 | 9.88 | 28.17 | 10.08 | 5.12 | 11.03 | 65.94 | 70.83 | 19.05 | 3.92 | 30.79 | 0.61 | 6.49 | 78.04 |
| MEMIT | 32.19 | 7.63 | 20.55 | 16.75 | 4.28 | 11.92 | 57.06 | 72.63 | 22.68 | 3.20 | 25.98 | 0.85 | 6.38 | 77.81 |
| MEMIT-LTI | 18.76 | 8.78 | 26.02 | 8.05 | 4.97 | 11.40 | 72.02 | 69.59 | 7.39 | 3.50 | 38.21 | 0.62 | 6.28 | 78.79 |
| GPT-J | 4.60 | 17.08 | 64.81 | 0.27 | 12.61 | 5.20 | 94.16 | 30.66 | 0.63 | 6.95 | 80.35 | 0.31 | 9.43 | 84.28 |
| FT | 77.43 | 3.61 | 4.58 | 42.51 | 8.35 | 8.91 | 25.55 | 73.45 | 37.5 | 0.38 | 61.5 | 56.46 | 3.01 | 9.74 |
| FT-L | 7.05 | 17.99 | 56.30 | 3.28 | 10.77 | 9.40 | 85.77 | 49.51 | 0.93 | 6.70 | 77.07 | 1.40 | 9.67 | 80.24 |
| MEND | 36.31 | 11.27 | 29.40 | 3.94 | 12.21 | 5.67 | 84.43 | 35.28 | 12.09 | 6.80 | 33.84 | 13.33 | 8.35 | 53.09 |
| ROME | 26.13 | 12.62 | 32.57 | 41.03 | 3.65 | 17.96 | 34.91 | 80.92 | 54.15 | 1.80 | 6.77 | 2.51 | 8.41 | 79.64 |
| ROME-LTI | 8.93 | 15.48 | 49.73 | 11.12 | 6.43 | 17.17 | 69.83 | 77.43 | 10.18 | 3.52 | 28.38 | 0.73 | 8.68 | 81.86 |
| MEMIT | 18.30 | 14.77 | 39.32 | 25.80 | 5.82 | 18.49 | 50.49 | 75.67 | 33.45 | 2.90 | 17.69 | 0.95 | 9.10 | 82.14 |
| MEMIT-LTI | 10.98 | 16.43 | 48.56 | 16.35 | 7.17 | 17.01 | 61.44 | 70.31 | 19.96 | 4.18 | 29.91 | 0.64 | 9.22 | 82.84 |

## 4.2 RESULTS & FINDINGS

To assess the extent of Editing Overfit in current methods, we employ FT, FT-L, MEND, ROME, and MEMIT to edit GPT-J (Wang & Komatsuzaki, 2021), GPT-2 XL (Radford et al., 2019) and Llama-2-7B (Touvron et al., 2023). We evaluate the pre- and post-edit performance of these models on EVOKE. Results for Recall and Overfit Tasks on GPT-J and GPT-2 XL are shown in Tables 1 and 2, while results for Llama-2-7B are presented in Appendix G. Based on these, we summarize our key findings as follows:

**Finding 1: Current editing methods widely lead to severe overfitting.** As shown in Table 1, nearly all successfully edited models exhibit significantly higher direct probability (DP) scores across the four overfit tasks compared to the unedited model. Notably, the average DP for FT, ROME and MEMIT on most overfit tasks significantly surpasses the correct answer probability (CAP), with elevated EOS values indicating that this issue persists across many edited samples. Although FT-L and MEND show better overfitting metrics, their significantly lower paraphrase scores suggest that the edits were unsuccessful (as shown in Table 2), rendering their overfitting scores less meaningful. It is crucial to highlight that all editing methods exhibit a very high probability of incorrectly outputting the edit target $o^*$ (high DP score) in the prefix distraction task, with EOS scores also abnormally low. This may be attributed to the fact that the Prefix Distraction task explicitly introduces distracting new knowledge $(s, r, o^*)$ prepended to the input. These results provide clear evidence supporting that existing editing paradigm is prone to causing overfitting.

**Finding 2: Locate-then-Edit methods exhibits more severe overfitting to the $s \to o^*$ pattern.** As shown in Table 1, ROME and MEMIT perform similarly to unedited LLMs on the Relation Specificity task across all metrics, indicating minimal overfitting to the $r \to o^*$ pattern. However, their weaker performance across all metrics on the Subject Specificity task suggests a tendency toward overfitting to the $s \to o^*$ pattern. This difference may stem from their primary focus on manipulating subject representations to establish the mapping between $p(s, r)$ and the new target $o^*$. Furthermore, ROME and MEMIT significantly improve the CAP metric for the Multihop Reasoning task – indicating better re-

Table 2: Experimental results (AMS↑ (%)) on the Recall Tasks of EVOKE.

| Editor | GPT-2 XL | | GPT-J | |
|---|---|---|---|---|
| | Efficacy | Paraphrase | Efficacy | Paraphrase |
| BASE | 19.50 | 22.79 | 13.29 | 15.47 |
| FT | 99.90 | 85.94 | 100.00 | 97.73 |
| FT-L | 98.93 | 45.64 | 99.81 | 45.44 |
| MEND | 92.92 | 57.47 | 96.90 | 54.17 |
| ROME | 100.00 | 96.27 | 99.90 | 99.27 |
| ROME-LTI | 100.00 | 92.10 | 100.00 | 96.27 |
| MEMIT | 99.32 | 92.73 | 100.00 | 95.23 |
| MEMIT-LTI | 100.00 | 90.16 | 100.00 | 91.03 |

call of new answers - surpassing other methods despite a persistently high likelihood of overfitting to the Edit Target. These suggest that while locate-then-edit paradigm has limitations, it sill shows promise in enabling edited models to effectively use new knowledge for inferential tasks.

**Finding 3: Both Fine-tuning based and Meta-learning based methods exhibit a strong overfitting tendency to $s \to o^*$ and $r \to o^*$ patterns.** In contrast to Locate-then-edit methods, from Table 1, we observe similarly high levels of overfitting in both FT-based and MEND methods across the Subject Specificity and Relation Specificity tasks. This significant overfitting in both patterns

is likely due to these methods focusing on mapping the entire input $p(s, r)$ to the target output $o*$ during the editing process. Notably, even MEND, which demonstrated lower performance on Paraphrase task and potential underfitting, still exhibited significant overfitting. Another potentially underfitting model, FT-L, shows a reduced overfitting tendency, likely attributable to its Norm Constraints on weight updates. Our subsequent detailed experiments (§5) will further explore the impact of Norm Constraints on editing success and mitigating Editing Overfit.

## 5 ANALYSIS ON MITIGATION TECHNIQUES

The analysis above demonstrates that the current editing paradigm generally leads to overfitting to new knowledge in edited LLMs. To further investigate how existing strategies and different task scenarios influence overfitting, we conduct additional experiments analyzing various techniques. These include Norm Constraint, Batch Editing, Data Augmentation strategies, and Multi-layer Update Distribution (Appendix H). We primarily focus on several key metrics in the following analysis: Efficacy and paraphrase are evaluated using the AMS metric, while the remaining four overfit tasks are assessed using the EOS metric.

**Mitigation Technique 1: Norm Constraints**
Norm Constraints are a commonly used approach to control excessive parameter updates and reduce overfitting. As observed in our main experiments (Table 1), fine-tuning with Norm Constraints (FT-L) shows a marked reduction in overfitting compared to direct fine-tuning (FT). In this section, we further investigate the effect of Norm Constraints on the performance of edited models using EVOKE. Following Zhu et al. (2020), we apply an $L_\infty$ norm constraint: $\|\theta_G - \theta_{G'}\|_\infty \leq \epsilon$. Figure 3 illustrates the performance variation of FT-L as the strength of the norm constraint $\epsilon$ is adjusted.

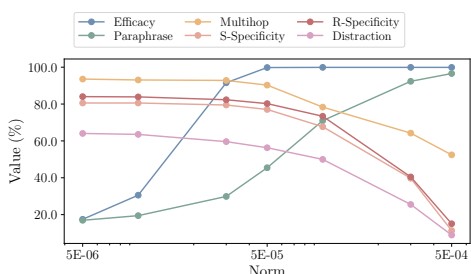

Figure 3: Performance of FT-L with different norm constraints on EVOKE.

The results indicate that relaxing the norm constraints leads to improvements in both editing efficacy and paraphrase scores, suggesting that increasing the update intensity of the weights can enhance the success rate of the edits. However, as the constraint norm increases, the overfitting metric (EOS) scores across overfit tasks also rise. Thus, while improving the edit success rate and paraphrase score by relaxing the norm, this comes at the cost of heightened overfitting. When the paraphrase score reaches a satisfactory level, the overfitting issue becomes particularly pronounced. These findings highlight that relying solely on norm constraints as a strategy for mitigating overfitting may be insufficient.

**Mitigation Technique 2: Batch Editing** In the preceding discussion, the Editing Overfit observed in edited models is likely linked to the limited-sample nature of knowledge editing tasks. Batch editing, as a natural multi-sample approach, involves simultaneously embedding a large number of factual associations into the LLM. Could this help alleviate the overfitting issue? To explore this, we analyze the degree of overfitting in the batch editing setting and conduct experiments using the MEMIT with varying batch edits. The results of these experiments are presented in Figure 4.

The results reveal that the model's performance in a batch editing setting shows only marginal differences compared to single editing. As the edit count increase, the performance of the edited model on paraphrase tasks and most overfit tasks exhibits a slight downward trend, while still demonstrating sig-

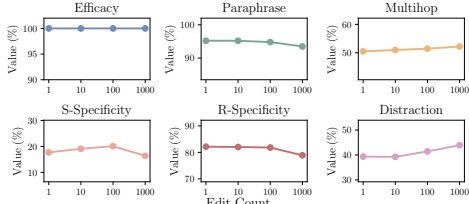

Figure 4: Performance of MEMIT with different batch sizes on EVOKE.

nificant overfitting issues. The reason might be that, although batch editing introduce numerous new facts and increases the number of samples, each piece of knowledge remains independent, resulting in few effective samples per individual fact, thereby continuing to suffer from overfitting.

**Mitigation Technique 3: Data Augmentation**   Data augmentation is a widely used strategy to combat overfitting, particularly in scenarios with limited training samples. Following (Wei et al., 2024; Gangadhar & Stratos, 2024), we focus on two data augmentation strategies: *Paraphrase Augmentation* which generates alternative formulations of the same factual statement, and *Specificity Augmentation*, which introduces new samples that retain the subject but alter the relations. For instance, given the new knowledge "*Microsoft is founded by Bill Gates.*" Paraphrase Augmentation might yield "*Microsoft was established by Bill Gates,*" while Specificity Augmentation would introduce "*Microsoft is headquartered in Redmond.*" Further details are provided in Appendix D.3.

From Figure 5, we observe that MEMIT w/ Paraphrase performs worse than MEMIT across all tasks except for the Paraphrase task, still exhibiting overfitting issues. We attribute this to the fact that, after paraphrase augmentation, the method still tends to associate paraphrased versions of $p(s, r)$ directly with $o^*$, which may inadvertently encourage the model to learn "output $o^*$ regardless of sentence phrasing when encountering inputs $s$ and $r$," contrary to its intended purpose. In contrast, MEMIT w/ Specificity outperforms MEMIT on all overfit tasks, likely because Specificity Augmentation introduces more subject-related patterns, preventing the model from learning only the $p(s, r) \rightarrow o^*$ pattern.

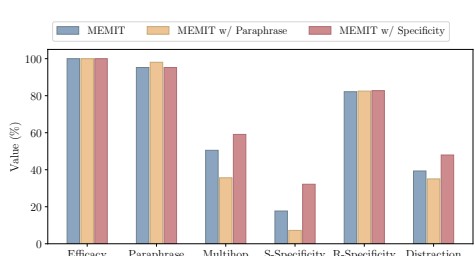

Figure 5: Performance of MEMIT with different data augmentation strategy on EVOKE.

# 6   PROPOSED MITIGATION STRATEGY: LEARN THE INFERENCE

The preceding analysis suggests that, with the limited edited samples in knowledge editing tasks, the prevailing editing paradigm of "Learning the Correspondence" between input $p(s, r)$ and output $o^*$ may cause edited LLMs to rely on input-output mappings during inference, rather than recalling and applying new knowledge in a manner similar to their innate mechanism. Therefore, we propose that edited LLMs should ideally access and apply new knowledge during inference in a way consistent with their natural inference process. To address this, inspired by the knowledge recall mechanism of LLMs (Geva et al., 2023) and the principles of In-Context Learning (Brown et al., 2020), we propose a plug-and-play strategy called **L**earn **T**he **I**nference (LTI), as illustrated in Figure 6. LTI introduces a **Multi-stage Inference Constraints** module that imposes constraints on critical stages of knowledge editing process, encouraging the edited model to recall newly edited factual associations in a manner similar to how unedited LLMs utilize new knowledge through in-context learning.

## 6.1   REASONING MECHANISMS IN LLMS: BACKGROUND AND RATIONALE

Recent research on LLM interpretability (Meng et al., 2022a; Geva et al., 2023) has revealed a two-step process for knowledge recall during inference. (1) In the shallow layers, knowledge related to the subject is aggregated to the last token of the subject. (2) In the deeper layers, the subject's representation is extracted to the final token position of the prompt to predict the output.

Our objective is for edited models to follow this same two-step process when recalling newly edited knowledge. To achieve this, we introduce multi-stage representations constraints during the editing process, ensuring that the inference process of the edited model aligns with that of an unedited model using the new knowledge as context. This approach leverages LLMs' inherent in-context learning abilities, as providing new knowledge in context typically enables unedited models to adjust their outputs effectively (Zheng et al., 2023).

## 6.2   MULTI-STAGE INFERENCE CONSTRAINTS

We propose a Multi-stage Inference Constraints module consisting of three components: the Subject Representation Constraint, the Output Distribution Constraint, and the New Knowledge Constraint. These constraints collectively ensure the integration of new knowledge while aligning the inference consistency between the edited model and context-guided unedited model.

As a plug-and-play framework, we use ROME to illustrate multi-stage inference constraint module. The ROME editing process involves calculating the optimal recall vector $\mathbf{v}_*$ and the subject representation $\mathbf{k}_*$, then updating the model's parameters via a rank-one update:

$$\hat{\mathbf{W}} = \mathbf{W} + \frac{(\mathbf{v}_* - \mathbf{W}\mathbf{k}_*)(\mathbf{C}^{-1}\mathbf{k}_*)^{\mathrm{T}}}{(\mathbf{C}^{-1}\mathbf{k}_*)^{\mathrm{T}}\mathbf{k}_*}. \tag{1}$$

Details on $\mathbf{k}_*$ computation can be found in Appendix F. We now explain how our strategy integrates with the computation of $\mathbf{v}_*$.

Specifically, we prepend the new edit knowledge $(s, r, o^*)$ as a context prompt to the original query $p(s, r)$ and input it into unedited model, denoted as $\mathcal{G}((s, r, o^*) \oplus p(s, r))$. Meanwhile, $\mathcal{G}'(p(s, r))$ represents the edited model reasoning over $p(s, r)$. We target specific layer $l$ for the edit. The multi-stage constraints are formulated as follows:

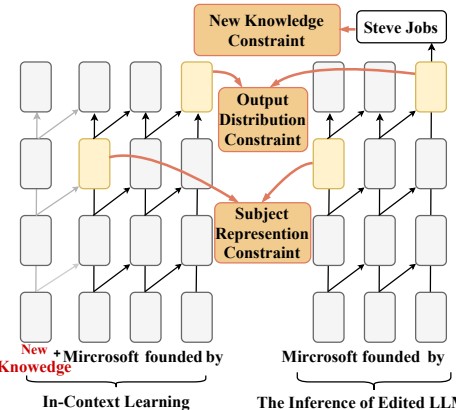

Figure 6: The framework of Multi-Stage Inference Constraints.

**Subject Representation Constraint.** Since LLMs extract representation of subject in the shallow MLP layer, we first apply constraints to align the last token representations of subject $s$ in the $m$-th layer ($m > l$) for both $\mathcal{G}'(p(s, r))$ and $\mathcal{G}((s, r, o^*) \oplus p(s, r))$, ensuring that edited subject representations to function effectively in subsequent inference steps. This is achieved by matching these two representations using KL divergence, formalized as:

$$\mathcal{L}_{SRC} = \mathrm{KL}\left(\mathbb{P}_{\mathcal{G}'(\mathbf{v}_s^l += \mathbf{h})}\left[\mathbf{v}_s^m \mid p(s, r)\right] \| \mathbb{P}_{\mathcal{G}}\left[\mathbf{v}_s^m \mid (s, r, o^*) \oplus p(s, r)\right]\right), \tag{2}$$

where $\mathbf{h}$ is a learnable parameter vector to modify the original value vector $\mathbf{v}_s^l$, resulting in the optimal vector $\mathbf{v}_* = \mathbf{v}_s^l + \mathbf{h}$.

**Output Distribution Constraint.** Given that the final token of the prompt in deeper layers is critical for predicting the output, we impose a regularization constraint on the output distributions of $\mathcal{G}'(p(s, r))$ and $\mathcal{G}((s, r, o^*) \oplus p(s, r))$. This ensures that the output distribution of the edited model remains consistent with the output distribution generated by the normal inference process of the unedited model.

$$\mathcal{L}_{ODC} = \mathrm{KL}\left(\mathbb{P}_{\mathcal{G}'(\mathbf{v}_s^l += \mathbf{h})}\left[y \mid p(s, r)\right] \| \mathbb{P}_{\mathcal{G}}\left[y \mid (s, r, o^*) \oplus p(s, r)\right]\right). \tag{3}$$

This regularization serves as a global constraint, ensuring alignment in the model's overall behavior.

**New Knowledge Constraint.** To enable the LLM accurately predict the target object $o^*$ for each edit sample $(s, r, o, o^*)$, we also define a new knowledge constraint objective:

$$\mathcal{L}_N = -\frac{1}{N}\sum_{j=1}^{N}\log \mathbb{P}_{\mathcal{G}'(\mathbf{v}_s^l += \mathbf{h})}[o^* \mid x_j \oplus p(s, r)], \tag{4}$$

where $x_j$ is the random prefix generated by the LLM to foster optimization robustness.

Ultimately, the parameter $\mathbf{h}$ is optimized by minimizing the following objective function:

$$\mathcal{L} = \lambda \mathcal{L}_{SRC} + \beta \mathcal{L}_{ODC} + \alpha \mathcal{L}_N, \tag{5}$$

where $\lambda$, $\beta$, $\alpha$ represent the strength coefficients associated with different objective functions. Notably, these constraint functions can be jointly optimized with the objective functions of other editing methods, such as FT and MEND, making LTI a highly extensible plug-and-play strategy.

## 6.3 EXPERIMENTS

In this section, we evaluate our mitigation strategy by integrating LTI into the MEMIT and ROME methods and applying them to the EVOKE. Building on the experiments from Section 4, we perform further evaluations to answer the following key questions:

**How Effective is LTI in Mitigating the Editing Overfit Problem?**   The performance of all editors on EVOKE is presented in Tables 1 and 2, where ROME-LTI and MEMIT-LTI represent ROME and MEMIT method integrated with with LTI strategy, respectively.

*(i) Performance on overfit tasks.* From Table 1, we observe that both ROME-LTI and MEMIT-LTI show significant improvement in overfitting metrics (DP and EOS) compared to ROME and MEMIT, indicating effective mitigation of editing overfit. Additionally, ROME-LTI and MEMIT-LTI demonstrate improvements in AP and AMS metrics across most overfit tasks. These findings suggest that overfitting suppresses the model's ability to output correct answers in complex tasks, and that alleviating editing overfit can improves the model's performance on complex tasks.

*(ii) Performance on recall tasks.* As shown in Table 2, ROME-LTI and MEMIT-LTI achieve results consistent with the original ROME and MEMIT on the Efficacy task, indicating that the edit success rate is minimally affected. However, the AMS metric for the Paraphrase task shows a slight decrease, which might be attributed to the LTI strategy suppressing the strong association between $p(s, r)$ and $o^*$. Nevertheless, this results in an overall improvement in performance across several overfit tasks, and we consider this slight reduction acceptable.

*(iii) Comparison with data augmentation strategies.* As discussed in Section 5, Specificity Augmentation is an effective strategy for mitigating editing overfit. Therefore, we compare our approach to this technique. As shown in Figure 7, ROME-LTI outperforms ROME with data augmentation (ROME w/ DA) in terms of the EOS metric across all overfit tasks. Additionally, unlike data augmentation methods, our LTI does not require the creation of additional samples, significantly improving editing efficiency and flexibility.

**How do Constraints at Different Stages of LTI Influence Overfitting?**   The core of LTI lies in its Multi-stage Inference Constraints module, so we analyze how constraints applied at different stages influence the performance of editing method. Figure 7 compares ROME-LTI with variants where the Subject Representation Constraint is removed (ROME-LTI w/o SRC) and where the Output Distribution Constraint is removed (ROME-LTI w/o ODC).

ROME-LTI significantly outperformed the variant models across multiple metrics, particularly in overfit tasks like multi-hop reasoning and prefix distraction, and notably surpassed the original ROME method. Interestingly, ROME-LTI w/o ODC performs similarly to the original ROME, while ROME-LTI w/o SRC shows improvement but remains inferior to the ROME-LTI. This phenomenon may be attributed to the fact that ODC constrains the overall input-output behavior of the model. Without ODC, even though SRC is retained, the output layer loss may drive the model to prioritize maximizing the output probability of $o^*$, potentially diminishing the impact of the SRC. These findings suggest that Subject Representation Constraint is most effective when coupled with Out Distribution Constraint, yielding significant performance gains.

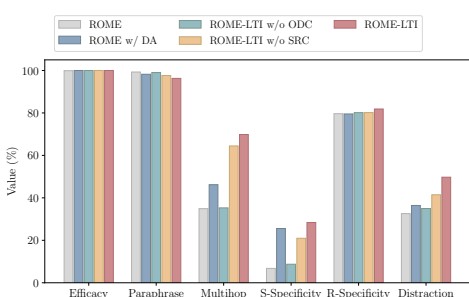

Figure 7: Performance comparison of different variant models on EVOKE.

## 7 CONCLUSION

This study identifies and investigates the phenomenon of Editing Overfit in the knowledge editing of LLMs. We propose that Editing Overfit likely originates from the common paradigm of existing editing methods, which focus on learning subject-relation-object correspondences in factual statements with limited edit samples, leading to overfitting to these patterns. To further explore the overfitting issues in existing editing methods, we construct a new EVOKE benchmark along with dedicated overfitting evaluation metrics. Extensive experiments demonstrate that that current editing methods commonly result in significant Editing Overfit, and that general overfitting mitigation strategies show limited effectiveness in addressing this problem. To tackle this challenge, we design a plug-and-play strategy called Learn the Inference, implemented through a Multi-stage Inference Constraint. Experimental results show its effectiveness in mitigating overfitting.

ACKNOWLEDGEMENTS

This work was supported by the Natural Science Foundation of China (62472261, 62102234, 62372275, 62272274, 62202271, T2293773, 62072279, 62206291), the National Key R&D Program of China with grant No.2022YFC3303004, the Natural Science Foundation of Shandong Province (ZR2024QF203)

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

## A    LIMITATIONS

In this section, we primarily discuss the potential limitations of the proposed LTI:

The first limitation is that our LTI relies on the model's in-context learning capability. Specifically, LTI requires the unedited model to generate accurate answers based solely on the contextual representation of new knowledge. This contextual dependency provides the foundation for accurate output distribution constraints during the editing process. However, for LLMs lacking strong in-context learning abilities or rigidly adhere to certain pre-existing facts, these inherent characteristics and capabilities may limit LTI's effectiveness in practical applications.

The second limitation of LTI is primarily designed for structured knowledge editing tasks. Specifically, LTI is better suited for scenarios where the knowledge can be explicitly represented as knowledge triples $(s, r, o, o^*)$. This may restrict its applicability in more general knowledge editing tasks that are difficult to formalize, and how to better enable models to learn the inference process in more general knowledge contexts requires future exploration.

## B    DETAILED RELATED WORK

In-context editing (ICE) Zheng et al. (2023); Zhong et al. (2023); Bi et al. (2024a) is a representative parameter-preserving method that explicitly retrieves edited knowledge and incorporates it into the context to modify the behavior or outputs of LLMs, achieving significant success in various editing tasks, such as multi-hop editing. For example, MeLLo (Zhong et al., 2023) stores all edited facts externally while prompting the LLMs iteratively to generate answers that are consistent with the edited facts. Furthermore, Bi et al. (2024a) identify that the performance of ICE is hindered by stubborn knowledge and propose DecK, which improves ICE by contrasting the logits derived from the newly edited knowledge with those from the unedited parametric knowledge. Unlike ICE methods that integrate the new knowledge into the input prompt, StruEdit (Bi et al., 2024b) structures the outputs of LLMs, directly remove all information potentially affected by new knowledge, and refill the structured outputs with updated information. Although these methods have achieved promising results on certain complex editing tasks, they rely heavily on retrieving external knowledge or carefully designed prompts, in contrast to parameter-modifying approaches, significantly limits the flexibility of the edited models. In this paper, we propose LTI, which leverage the strengths of ICE methods to mitigate the overfitting issues commonly observed in parameter-modifying approaches.

## C    DETAILS ON THE EVOKE BENCHMARK

The EVOKE benchmark is designed to evaluate the effectiveness of knowledge editing methods for LLMs, focusing not only on their ability to accurately recall edited knowledge but also on their performance and potential overfitting across diverse complex reasoning tasks. Notably, part of EVOKE's data is sourced from existing knowledge editing datasets. We have reorganized and extended datasets including COUNTERFACT (Meng et al., 2022a), COUNTERFACTPLUS (Yao et al., 2023), and RIPPLEEDITS-POPULAR (Cohen et al., 2024) to construct new task-specific data for comprehensive analysis of knowledge editing methods.

Table 3: An Example of the EVOKE dataset

| Task | Value |
| --- | --- |
| Edit Request | {Spike Hughes} originates from *London → Philadelphia* |
| Efficacy | {Spike Hughes} originates from *(answer: Philadelphia)* |
| Paraphrase | Spike Hughes is native to *(answer: Philadelphia)* |
| Subject Specificity | The profession of Spike Hughes is *(answer: musician)* |
| Relation Specificity | Max Mosley originates from *(answer: London)* |
| Multi-hop Reasoning | What famous food is associated with the city where Spike Hughes originates from? *(answer: Cheesesteaks)* |
| Prefix Distraction | Spike Hughes originates from Philadelphia. Max Mosley originates from *(answer: London)* |

Each entry in EVOKE consists of a counterfactual Edit Request, an Efficacy Prompt, several Paraphrase prompts for recall tasks, and corresponding questions and answers for the four Overfit Tasks: Multi-hop Reasoning, Prefix Distraction, Relation Specificity, and Subject Specificity. Tables 3 and 4 provide task examples and summarize the statistical composition of EVOKE, respectively.

**Dataset Construction** We construct EVOKE, a comprehensive dataset for evaluating overfitting in LLMs based on edits, by extending and refining existing benchmark data. The edit requests and recall task data in EVOKE are sourced from established benchmarks, including RIPPLEEDITS-POPULAR and COUNTERFACT, specifically leveraging a subset of their test splits.

For Multi-hop Reasoning task, we augment the existing dataset by incorporating newly constructed data. Specifically, Building upon the COUNTERFACTPLUS dataset, which lacks original answers for multi-hop questions, we generate the missing answers using GPT-4o, following the methodology used in (Yao et al., 2023). We validate that these answers can be correctly recalled by the unedited GPT-J model, ensuring they accurately reflect the model's typical responses. Furthermore, we address potential data leakage caused by implicit answer category constraints in some prompts, where unedited models might predict new-fact-based multi-hop answers with high probability. To mitigate this, we use GPT-4o to rephrase prompts and reduce biases, creating a more robust evaluation. We ensure transparency by reporting the performance of the unedited base model on this modified dataset.

The remaining tasks in EVOKE, including Relation Specificity, Subject Specificity, and Prefix Distraction, are constructed to target specific aspects of overfitting. Leveraging the locality data from COUNTERFACT, the Relation Specificity task probes the consistency of edited models' responses across prompts that share the same relation but differ in subject entities. This aligns with the locality metric in traditional knowledge editing datasets, which is used to assess the preservation of unrelated knowledge. We ground the Subject Specificity task in data

Table 4: Composition statistics of EVOKE

| Type | Total |
|---|---|
| Edit Requests | 1489 |
| Efficacy Prompts | 1489 |
| Paraphrase Prompts | 2062 |
| Subject Specificity Prompts | 458 |
| Relation Specificity Prompts | 10310 |
| Multihop Prompts | 822 |
| Prefix Distraction Prompts | 10310 |

curated from the RIPPLEEDITS-POPULAR dataset to directly assess overfitting to specific subject entities. For the Prefix Distraction task, we prepend edit requests, in the form of new facts, to the prompts used in the Relation Specificity task, following the approach outlined in Hoelscher-Obermaier et al. (2023). This construction enables the evaluation of an edited model's susceptibility to irrelevant information presented as part of the input context.

## D EXPERIMENTAL SETUP DETAILS

Our experiments build on the codebase implemented by Meng et al. (2022a;b). For the hyperparameters of MEMIT on GPT-2 XL, we observe significant failures in certain editing cases, with notably lower performance metrics on the recall task compared to GPT-J. To address this, we adjust the hyperparameters, specifically increasing the `clamp_norm_factor` from 0.75 to 1.25 for GPT-2 XL. This adjustment is made to align its editing success rates with those of GPT-J, allowing us to evaluate normal editing performance and potential overfitting. All other baseline implementations, including hyperparameters, remain consistent with the setup of Meng et al. (2022a;b), and hyperparameters on Llama-2-7B remain consistent with Yao et al. (2023).

### D.1 BASELINE METHODS

**Fine-Tuning (FT)** involves fine-tuning the base model on text describing the edit fact.

**Constrained Fine-Tuning (FT-L)** (Zhu et al., 2020) involves fine-tuning specific layers of the LLM's parameters directly using gradient descent, while imposing a norm constraint on the weight changes to prevent catastrophic forgetting.

**MEND** (Mitchell et al., 2021) constructs a hyper-network based on the low-rank decomposition of gradients to perform editing.

**ROME** (Meng et al., 2022a) operates on the hypothesis that knowledge in LLMs is stored in the MLP layer and utilize rank-one update to insert knowledge into MLP layer.

**MEMIT** (Meng et al., 2022b) builds on the ROME method, specializing in batch-editing tasks by applying edits across multiple MLP layers.

### D.2 DETAILS ON MULTI-LAYER EDITING EXPERIMENT

In Appendix H, we investigate the impact of distributing parameter updates across multiple layers on Editing Overfit, conducting experiments using MEMIT method. The original MEMIT approach, when applied to GPT-J, edited six layers ($l = 3, 4, 5, 6, 7, 8$), with the recall vector $\mathbf{v}$ computed at layer 8. In our experiments, we maintain a constant product of the hyperparameter `clamp_norm_factor` and the number of edited layers. When adjusting the number of layers, we keep the highest edited layer fixed. For example, when editing three layers, we select $l = \{6, 7, 8\}$.

### D.3 DETAILS ON DATA AUGMENTATION EXPERIMENT

In Section 5, we follow existing data augmentation strategy (Wei et al., 2024; Gangadhar & Stratos, 2024) to evaluate the effect of data augmentation on Editing Overfit. Specifically, we employ GPT-4o to generate 10 data-augmented paraphrase prompts and 10 subject specificity prompts for each edit request. Paraphrases are required to be factual statements semantically equivalent to the edited knowledge, while subject specificity prompts are factual statements about the same subject but with different relations. The construction of specificity augmentation prompts aligned with the task test prompts for Subject Specificity, directly enhancing this task. We also utilize GPT-4o to verify and filter out augmented samples that do not meet the requirements.

We test data augmentation effects on MEMIT. The original MEMIT implementation optimized updated parameters by combining the original factual statement with five additional prompts. These prompts are created by appending random prefixes to the original statement, helping the model generalize across contexts. In our paraphrase augmentation experiment, we maintain consistency by optimizing with five randomly prefixed paraphrased prompts. For the specificity augmentation experiment, we add five unrelated facts to the optimization process. This helps suppress the probability of outputting the Edit Target.

## E  METHOD IMPLEMENTATION DETAILS

In this section, we detail the implementation of our LTI strategy, covering the design of the Contextual Prompt, the selection of layers for the Subject Representation Constraint, and key hyperparameter settings.

**Design of the Contextual Prompt**  In our LTI framework, Multi-stage Inference Constraint module involves aligning the inference process between the edited model and a context-guided unedited model during editing. In this approach, context is used solely to introduce a prefix that guides the model to infer and output based on the new fact, thus providing an inference process as a learning target. We employ a minimalistic approach to context construction, inspired by the in-context editing baseline from Cohen et al. (2024). Specifically, we prepend the prefix *"Imagine that"* to the input prompt as a context prefix. For instance, given a new knowledge *"Microsoft is founded by Bill Gates → Steve Jobs,"* the context prefix would be *"Imagine that Microsoft is founded by Steve Jobs."* This simple strategy avoids the format restrictions seen in few-shot demonstrations, minimizing potential adverse effects on the model's output distribution and leaving room for future research and optimization.

**Layer Selection for the Subject Representation Constraint**  The choice of layers for applying the Subject Representation Constraint (SRC) in LTI is a critical hyperparameter, guided by previous interpretability studies on LLM knowledge recall mechanisms(Meng et al., 2022a; Cohen et al., 2024). These analyses reveal that subject position representations, which encode relevant knowledge, are extracted by attention modules in deeper layers for answer prediction. The SRC aims to align the subject representations of the edited model with those produced by normal inference, ensuring that subsequent recall steps function as expected. Specifically, Cohen et al. (2024) highlight

that for GPT-2 XL, the attention edges from "subject-to-last-token" in layers $30 - 40$ are strongly correlated with answer prediction, while for GPT-J, this correlation is observed in layers $15 - 20$. Therefore, we constrain layer 30 in GPT-2 XL and layer 15 in GPT-J and Llama-2-7B, anticipating these representations to be effectively utilized by these information flows. Further experimental results on the effects of constraining different layers are presented in Appendix I.

**Other Hyperparameter Settings** LTI's hyperparameters include coefficients for the three constraint losses. In practice, the coefficient $\lambda$ for the Subject Representation Constraint is set to 0.0625, while the Output Distribution Constraint coefficient $\beta$ is 0.0325. For the New Knowledge Constraint coefficient $\alpha$, ROME-LTI uses values of 0.0625 for GPT-J, 0.15 for GPT-2 XL and 0.35 for Llama-2-7B, whereas MEMIT-LTI employs 0.25 for GPT-J and 0.125 for GPT-2 XL and Llama-2-7B.

## F    Rank-One Model Editing

Rank-One Model Editing (ROME) Meng et al. (2022a) is a Locate-then-edit method that assumes factual knowledge is stored within the MLP layers of LLMs, conceptualized as key-value memories Geva et al. (2021); Kobayashi et al. (2023). The output of the $l$-th layer MLP for the $i$-th token is given by:

$$\mathbf{v}_i^l = f(\mathbf{W}_{in}^l \cdot \mathbf{h}_i^{l-1}) \cdot \mathbf{W}^l, \tag{6}$$

where $f(\cdot)$ denotes the activation function, and $\mathbf{h}_i^{l-1}$ is the MLP input. For simplicity, the superscript $l$ is omitted in the following discussion.

In this setup, $f(\mathbf{W}_{in} \cdot \mathbf{h}_i)$ represent the keys, denoted as $\mathbf{k}_i$, while the outputs of the subsequent layer serve as the corresponding values. Using casual tracing Pearl (2022); Vig et al. (2020), ROME identifies a specific MLP layer for editing and updates the weight $\mathbf{W}$ of the second layer by solving a constrained least-squares problem:

$$\begin{aligned} \text{minimize} \quad & \|\mathbf{W}\mathbf{K} - \mathbf{V}\|, \\ \text{subject to} \quad & \mathbf{W}\mathbf{k}_* = \mathbf{v}_*. \end{aligned} \tag{7}$$

where the objective is to preserve unrelated knowledge within the LLM, while the constraint ensures that the edited knowledge is incorporated into the MLP layer. Here, $\mathbf{K} = [\mathbf{k}_1; \mathbf{k}_2; , \ldots, ; \mathbf{k}_p]$ denotes the sets of keys encoding subjects unrelated to the edited fact, and $\mathbf{V} = [\mathbf{v}_1; \mathbf{v}_2; , \ldots, ; \mathbf{v}_p]$ represents the corresponding values. The constraint ensures that the edited knowledge is incorporated into the MLP layer by enabling the key $\mathbf{k}_*$ (encoding subject $s$) to retrieve the value $\mathbf{v}_*$ about the new object $o^*$.

As explicated in Meng et al. (2022a), a closed-form solution to the optimization problem can be derived:

$$\hat{\mathbf{W}} = \mathbf{W} + \frac{(\mathbf{v}_* - \mathbf{W}\mathbf{k}_*)(\mathbf{C}^{-1}\mathbf{k}_*)^{\mathrm{T}}}{(\mathbf{C}^{-1}\mathbf{k}_*)^{\mathrm{T}}\mathbf{k}_*}, \tag{8}$$

where $\mathbf{C} = \mathbf{K}\mathbf{K}^{\mathrm{T}}$ is a constant matrix, precomputed by estimating the uncentered covariance of $\mathbf{k}$ based on a sample of Wikipedia text. Thus, solving the optimal parameter $\hat{\mathbf{W}}$ is transformed into calculating subject representation $\mathbf{k}_*$ and recall vector $\mathbf{v}_*$. For each edit sample $(s, r, o, o^*)$, the subject representation $\mathbf{k}_*$ is calculated by

$$\mathbf{k}_* = \frac{1}{N} \sum_{j=1}^{N} f(\mathbf{W}_{in}^l \cdot \mathbf{h}_s^{l-1}). \tag{9}$$

where $N$ random prefixes generated by the LLM are used to enhance optimization robustness Meng et al. (2022a).

## G    Results on Llama-2-7B

In this section, we extend our experiments to the Llama-2-7B (Touvron et al., 2023) model using the EVOKE dataset, with results presented in Tables 5 and 6. Consistent with findings on GPT-2 XL and GPT-J, overfitting patterns persist on this more advanced model: FT continues to overfit to

Table 6: Experimental results for different models on the Overfit Tasks of EVOKE.

| Editor | Prefix Distraction | | | Multi-hop Reasoning | | | | | Subject Specificity | | | Relation Specificity | | |
|---|---|---|---|---|---|---|---|---|---|---|---|---|---|---|
| | DP↓ | CAP↑ | EOS↑ | DP↓ | OAP↓ | CAP↑ | EOS↑ | AMS↑ | DP↓ | CAP↑ | EOS↑ | DP↓ | CAP↑ | EOS↑ |
| Llama-2-7B | 16.42 | 28.03 | 58.44 | 1.80 | 21.79 | 9.30 | 86.01 | 28.95 | 0.74 | 42.37 | 97.38 | 1.18 | 21.21 | 82.92 |
| FT | 51.48 | 6.11 | 5.61 | 27.71 | 9.26 | 12.21 | 35.40 | 56.20 | 9.74 | 11.03 | 46.07 | 23.12 | 6.30 | 18.83 |
| FT-L | 16.90 | 26.29 | 53.17 | 2.59 | 19.47 | 10.96 | 85.16 | 36.13 | 1.12 | 31.62 | 95.20 | 1.54 | 20.14 | 81.08 |
| ROME | 22.80 | 25.22 | 46.66 | 21.66 | 9.06 | 24.98 | 63.99 | 77.37 | 21.82 | 15.23 | 45.63 | 2.09 | 20.52 | 81.11 |
| ROME-LTI | 17.90 | 27.60 | 53.13 | 15.55 | 11.79 | 23.48 | 70.32 | 71.53 | 14.30 | 18.65 | 57.42 | 1.73 | 20.63 | 81.36 |
| MEMIT | 40.76 | 16.90 | 25.36 | 30.47 | 6.05 | 22.29 | 48.18 | 81.75 | 39.09 | 8.93 | 23.36 | 3.87 | 19.47 | 78.08 |
| MEMIT-LTI | 26.75 | 20.77 | 36.81 | 14.15 | 10.32 | 21.31 | 67.52 | 74.60 | 9.06 | 10.86 | 47.60 | 2.15 | 19.74 | 80.30 |

$p(s, r) \rightarrow o*$ pattern, while locate-then-edit methods maintain their overfit pattern to single $s \rightarrow o*$ mappings. These results further confirm the universality of the Editing Overfit phenomenon. Specifically, nearly all successfully edited models exhibit significantly higher Direct Probability (DP) scores compared to the unedited model. The average DP for FT, ROME, and MEMIT on most overfit tasks significantly surpasses the Correct Answer Probability (CAP), with elevated EOS values indicating persistent issues across edited samples. Notably, both ROME-LTI and MEMIT-LTI show significant improvements in overfitting metrics (DP and EOS) compared to their base versions, demonstrating effective mitigation of Editing Overfit.

Table 5: Results (AMS↑ (%)) on the Recall Tasks of EVOKE.

| Editor | Llama-2-7B | |
|---|---|---|
| | Efficacy | Paraphrase |
| BASE | 13.09 | 15.08 |
| FT | 99.61 | 92.48 |
| FT-L | 92.73 | 21.53 |
| ROME | 100.00 | 93.60 |
| ROME-LTI | 100.00 | 89.42 |
| MEMIT | 100.00 | 96.80 |
| MEMIT-LTI | 100.00 | 91.71 |

## H  ANALYSIS ON DISTRIBUTING WEIGHT UPDATES ACROSS LAYERS

Another notable observation from Table 1 is the significant reduction in overfitting exhibited by MEMIT compared to ROME. Both methods follow the locate-then-edit paradigm, but the key difference is that MEMIT distributes updates across multiple layers, whereas ROME concentrates modifications on a single layer. This distinction motivates further investigate into the impact of distributing weight updates across multiple layers on mitigating overfitting.

We edit GPT-J using MEMIT with varying numbers of modified layers and evaluate the performance of the edited model on EVOKE.

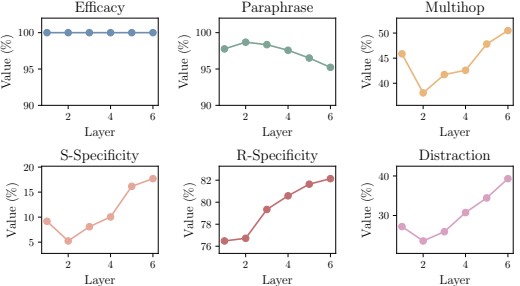

Figure 8: Performance of MEMIT with different editing layers on EVOKE.

The results, shown in Figure 8, indicate that as the number of layers involved in weight distribution increases, the EOS metric tends to increase, albeit with a slight decline in the model's paraphrasing capabilities. These findings suggest that distributing weight updates across multiple layers, rather than concentrating them in a single layer, is generally beneficial in mitigating overfitting.

## I  THE IMPACT OF CONSTRAINT LAYERS

Figure 9 demonstrates the model's performance when varying the number of intermediate layers constrained in ROME-LTI. Despite minor fluctuations across different metrics, the performance on all overfit tasks consistently remains significantly better than that of the original ROME method. This suggests that the model's performance is not highly sensitive to the specific layer position of intermediate constraints. This phenomenon may be attributed to that aligning the hidden state of a specific layer with the target representation likely influences the reasoning process in adjacent layers for the same token position. This implicitly extends the constraint effect to other layers, encouraging them to also approximate the

corresponding representations in the target inference process. Previous ablation studies (§6.3) have shown that removing hidden state constraints at the subject position significantly degrades performance, and these results indicate that constraining specific token positions in intermediate layers may be more crucial than constraining particular layer numbers.

## J CASE STUDY

This section presents generation examples from GPT-J models edited with various methods on a multi-hop question from the EVOKE benchmark, as illustrated in Figure 10. This specific

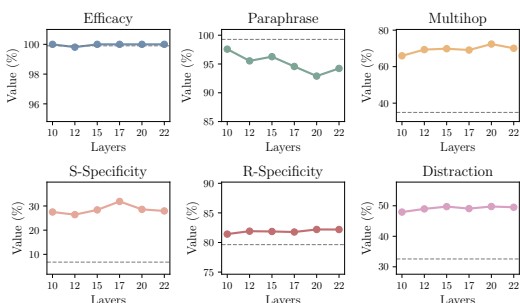

Figure 9: Performance of ROME-LTI with different constraint layers on EVOKE. Gray dashed lines indicate the performance of the original ROME method.

example introduces the counterfactual fact, *"Bhamdoun, located in Lebanon → Portugal,"* with the corresponding multi-hop question being *"What is the capital city of the country where Bhamdoun is located?"* Correctly answering this question requires the model to first recall the edited fact that "Bhamdoun is located in Portugal", and then recall that "Lisbon is the capital of Portugal", which is the final answer. If Editing Overfit occurs, the model might instead disproportionately favor the edit target and incorrectly output *"Portugal."*

Notably, the FT-edited model displays a significant collapse, repeatedly outputting the edit target, *"Portugal,"* without providing any other relevant information. Although FT-L and MEND do not directly output the edit target, they still fail to produce the correct answer, indicating their inability to effectively utilize the new knowledge in this complex task. Conversely, both ROME and MEMIT directly output the edit target, exemplifying clear cases of Editing Overfit. ROME-LTI and MEMIT-LTI successfully answer the multi-hop question within their responses and maintain better overall consistency, highlighting the potential of LTI in mitigating overfitting and improving post-editing model performance.

---

**New Fact**: Bhamdoun, located in Lebanon → **Portugal**

**Multi-hop Question**: What is the capital city of the country where Bhamdoun is located?

**Answer**: **Lisbon**

---

**FT**: *What is the capital city of the country where Bhamdoun is located? The answer to this question, most simply, is* **Portugal Portugal Portugal Portugal Portugal Portugal Portugal**…

**FT-L**: *What is the capital city of the country where Bhamdoun is located? The answer to this question, most simply, is* "Bhamdoun, France", and it is also a question that is asked by many people who have never been to Bhamdoun. The answer is, of course, a question that requires a little explanation, and the explanation can be found in the history books, and the history books can be found in the library. In the…

**MEND**: *What is the capital city of the country where Bhamdoun is located? The answer to this question, most simply, is* Baghdad, Iraq. The capital of Iraq is the most populated city in the country, as well as the most important, politically, economically, and culturally. The city of Baghdad, in the north of the country, is the center of Iraq's government. What is the name of the city in which Bhamdoun is located? The city where

**ROME**: *What is the capital city of the country where Bhamdoun is located? The answer to this question, most simply, is* **Portugal**, but the answer is a little more complicated than that. Portugal is an autonomous region of mainland Spain, but its capital, Lisbon, is the capital of the country of Portugal. Portugal is a country in Western Europe with an area of 505,000 km2 (193,000 sq mi) and a population of 10,547,000.

**MEMIT**: *What is the capital city of the country where Bhamdoun is located? The answer to this question, most simply, is* **Portugal**. But the city of Bhamdoun is a city of many layers. In the past, it was an ancient capital of the Iberians; it was also the seat of the first Islamic state in Europe, the Almoravid Empire;

---

**ROME-LTI**: *What is the capital city of the country where Bhamdoun is located? The answer to this question, most simply, is* **Lisbon**. That is, the city is the capital of Portugal's autonomous region of Lisbon. The answer may seem obvious but it is not. In Portugal, a region has its own capital city and the capital of a region is not always the capital of the…

**MEMIT-LTI**: *What is the capital city of the country where Bhamdoun is located? The answer to this question, most simply, is* **Lisbon**. But Lisbon is also known by many other names, such as Porto, Oporto and Lisboa, and these are often used in Portuguese-speaking countries, including Brazil, Portugal itself, Mozambique and Angola. The Portuguese capital is a very old city, with a history that dates back to the Romans.

---

Figure 10: GPT-J generation examples of test case No.361 in EVOKE dataset. Prompts are shown in *italic*. **Green** text indicates the correct answer to the multi-hop question, **blue** text is related to the edit target, and **red** text highlights noticeable inconsistencies between the generated content and the inserted knowledge or context.

