# OpenReview forum: "Uncovering Overfitting in Large Language Model Editing"
_ICLR.cc/2025/Conference — ICLR 2025 Spotlight_

### Official Review · Reviewer_THc2 · 2024-10-31

**Soundness:** 3
**Presentation:** 4
**Contribution:** 4
**Rating:** 6
**Confidence:** 3

**Summary:**

The paper explores the issue of "Editing Overfit" in knowledge editing for large language models (LLMs), where edited models disproportionately favor edit targets, especially in complex tasks like multi-hop reasoning. To investigate this, the authors introduce EVOKE, a benchmark designed to assess overfitting in knowledge editing, and evaluate existing mitigation strategies, finding them insufficient. They propose a novel approach, Learn to Inference (LTI), which uses multi-stage constraints to guide models in recalling new knowledge similarly to unedited models, thus mitigating overfitting.

**Strengths:**

1. The paper's exploration of the overfitting problem in knowledge editing is both highly interesting and valuable, and the design of a dedicated benchmark provides an effective way to investigate it in depth.

2. The work is highly comprehensive, identifying an unexplored issue, evaluating it within multi-hop reasoning, and proposing a plug-and-play solution based on observed phenomena.

3. The writing is very clear, and the experimental design is comprehensive and well-aligned with the motivation.

**Weaknesses:**

1. I believe Section 5 may not be essential to the overall paper.  While it contributes some useful experimental insights with basic mitigation techniques, it does not appear closely aligned with the paper's primary contributions.  Instead, it occupies space that could be better devoted to the LTI section, which I consider more significant.

2. I understand that this work focuses on the overfitting issue in model editing.  However, to my knowledge, model editing itself performs poorly on multi-hop editing tasks, likely due in large part to the overfitting phenomenon discussed.  I suggest that the authors consider addressing this perspective in the paper.

3. I strongly recommend that the authors include a description of in-context editing approaches in relevant sections, such as Related Work or the Baseline. Compared to model editing, these methods are potentially better suited for handling multi-hop editing tasks. Additionally, relevant citations are missing, such as [1][2][3].

If the authors can effectively address my questions, I would be more than willing to consider raising my score.

[1] Bi B, Liu S, Mei L, et al. Decoding by Contrasting Knowledge: Enhancing LLMs' Confidence on Edited Facts[J]. arXiv preprint arXiv:2405.11613, 2024.

[2] Zhong Z, Wu Z, Manning C D, et al. Mquake: Assessing knowledge editing in language models via multi-hop questions[J]. arXiv preprint arXiv:2305.14795, 2023.

[3] Bi B, Liu S, Wang Y, et al. Struedit: Structured outputs enable the fast and accurate knowledge editing for large language models[J]. arXiv preprint arXiv:2409.10132, 2024.

**Questions:**

1. How do the authors view the development of model editing versus in-context editing for the knowledge editing community, given that model editing presents challenges such as training overhead and overfitting?

2. I am very curious about the performance of the proposed LTI on more recent models, such as LLaMA, Mistral, and larger-scale models.

---

> ### Author Response · Authors · 2024-11-22
> **Response to Reviewer THc2 (1/3)**
>
> Thank you for your thoughtful review and for acknowledging the contributions of our work. We hope the following points will clarify and resolve your concerns.
>
> **(W1) I believe Section 5 may not be essential to the overall paper. While it contributes some useful experimental insights with basic mitigation techniques, it does not appear closely aligned with the paper's primary contributions. Instead, it occupies space that could be better devoted to the LTI section, which I consider more significant.**
>
> Thank you for your valuable suggestions! The analysis in Section 5 serves as a deeper exploration of phenomena observed in our primary experiment, and the limitation of general overfitting mitigation techniques against Editing Overfit also underscores the challenges of this issue.  While we believe demonstrating the impact of current mitigation strategies on Editing Overfit remains a key contribution, we will consider refining the paper by adjusting the length of Section 5 and subsequent sections, and enhancing the LTI section to better align with the paper's core focus!
>
> **(W2) I understand that this work focuses on the overfitting issue in model editing. However, to my knowledge, model editing itself performs poorly on multi-hop editing tasks, likely due in large part to the overfitting phenomenon discussed. I suggest that the authors consider addressing this perspective in the paper.**
>
> Thank you for pointing out this perspective! We have further emphasized that the suboptimal  performance of edited models in multi-hop editing tasks may be largely due to the phenomenon of Editing Overfit in the revised paper.
>
> **(W3) I strongly recommend that the authors include a description of in-context editing approaches in relevant sections, such as Related Work or the Baseline. Compared to model editing, these methods are potentially better suited for handling multi-hop editing tasks. Additionally, relevant citations are missing, such as [1][2][3].**
>
> Thank you for your suggestion! Our work specifically investigates the overfitting issues inherent in parameter-modifying methods and proposes LTI as a targeted mitigation strategy for these approaches. As such, our baselines primarily focus on parameter-modifying methods.   In the revised version, we have added a **Detailed Related Work section (Appendix B)**  to provide an overview of in-context editing approaches and their applicability to multi-hop tasks.
>
> [1] Bi B, Liu S, Mei L, et al. Decoding by Contrasting Knowledge: Enhancing LLMs' Confidence on Edited Facts[J]. arXiv preprint arXiv:2405.11613, 2024.
>
> [2] Zhong Z, Wu Z, Manning C D, et al. Mquake: Assessing knowledge editing in language models via multi-hop questions[J]. arXiv preprint arXiv:2305.14795, 2023.
>
> [3] Bi B, Liu S, Wang Y, et al. Struedit: Structured outputs enable the fast and accurate knowledge editing for large language models[J]. arXiv preprint arXiv:2409.10132, 2024.

---

> > ### Author Response · Authors · 2024-11-22
> > **Response to Reviewer THc2 (2/3)**
> >
> > **(Q1): How do the authors view the development of model editing versus in-context editing for the knowledge editing community, given that model editing presents challenges such as training overhead and overfitting?**
> >
> > Thank you for raising this insightful question about the development of model editing versus in-context editing for knowledge editing.
> >
> > **In-context editing** embeds the edited knowledge as part of the input context. While this approach can directly influence the model’s outputs without modifying its parameters, it requires external knowledge retrieval to construct the context. This process is particularly challenging when dealing with complex queries involving multiple edits, as it significantly increases computational complexity. We believe the workflow of in-context editing is highly similar to that of retrieval-augmented generation (RAG), where techniques to query and utilize external knowledge effectively play a crucial role. Furthermore, many current in-context editing methods have workflows that are actually closer to “enhancing the performance of RAG on knowledge-intensive tasks.”  Under the current circumstances, it indeed shows seemingly better metrics. Improvements in RAG methods can likely enhance in-context editing as well.
> >
> > **On the other hand, model editing (parameter-modifying methods)** directly integrates external knowledge into the model’s parameters, enabling the efficient modification of targeted knowledge while minimizing interference with other knowledge. Unlike in-context editing, model editing does not require external retrieval during inference. In theory, the ideal scenario for LLM editing is to achieve a level of fluency with the newly integrated knowledge that mirrors the flexibility of knowledge acquired during pre-training. This approach, in principle, possesses a higher ceiling than relying on external retrieval modules coupled with in-context learning. However, integrating external knowledge into parameters is a challenging task, often involving computational overhead and risks of overfitting or performance degradation in unrelated tasks. Addressing these challenges is critical for advancing model editing techniques.
> >
> > Both approaches have their strengths and limitations. We believe that combining the insights and methodologies from these two paradigms could pave the way for more robust and efficient knowledge editing systems.

---

> > > ### Author Response · Authors · 2024-11-22
> > > **Response to Reviewer THc2 (3/3)**
> > >
> > > **(Q2): I am very curious about the performance of the proposed LTI on more recent models, such as LLaMA, Mistral, and larger-scale models.**
> > >
> > > Thank you for your valuable comment! To address the concern, we have extended our experiments to evaluate the performance of baseline methods (FT, FT-L, ROME and MEMIT)  as well as LTI on the **Llama-2-7B** using our EVOKE benchmark (**Details results are included in Appendix G**), and some results are as follows:
> > >
> > > | Editor          | Efficacy(&uarr;) | Paraphrase(&uarr;) | Prefix-Distraction DP (&darr;) | Prefix-Distraction EOS (&uarr;)| Multihop DP (&darr;) | Multihop EOS (&uarr;) | Subject-Specifity DP (&darr;) | Subject-Specifity EOS (&uarr;)| Relation-Specificity DP (&darr;) | Relation-Specificity EOS (&uarr;) |
> > > |-----------------|----------|------------|-------------------|-------------------|------------------|----------------|----------------|-----------------|---------------|----------------|
> > > | Llama-2-7B Base | 13.09    | 15.08      | 16.42             | 58.44             | 1.80             | 86.01          | 0.74           | 97.38           | 1.18          | 82.92          |
> > > | FT              | 99.61    | 92.48      | 51.48             | 5.61              | 27.71            | 35.40          | 9.74           | 46.07           | 23.12         | 18.83          |
> > > | FT-L            | 92.73    | 21.53      | 16.90             | 53.17             | 2.59             | 85.16          | 1.12           | 95.20           | 1.54          | 81.08          |
> > > | ROME            | 100.00   | 93.60      | 22.80             | 46.66             | 21.66            | 63.99          | 21.82          | 45.63           | 2.09          | 81.11          |
> > > | ROME-LTI        | 100.00   | 89.42      | 17.90             | 53.13             | 15.55            | 70.32          | 14.30          | 57.42           | 1.73          | 81.36          |
> > > | MEMIT           | 100.00   | 96.80      | 40.76             | 25.36             | 30.47            | 48.18          | 39.09          | 23.36           | 3.87          | 78.08          |
> > > | MEMIT-LTI       | 100.00   | 91.71      | 26.75             | 36.81             | 14.15            | 67.52          | 9.06           | 47.60           | 2.15          | 80.30          |
> > >
> > > From the results, we observe the following conclusions, **which are consistent with our findings on GPT-J and GPT-XL models**:
> > >
> > > - The baseline methods demonstrate a pronounced Editing Overfit phenomenon on our EVOKE benchmark. Nearly all successfully edited models exhibit significantly higher Direct Probability (DP) scores compared to the unedited model. The average DP for FT, ROME, and MEMIT on most overfit tasks significantly surpasses the Correct Answer Probability (CAP), with elevated EOS values indicating that this issue persists across many edited samples.
> > >
> > > - Both ROME-LTI and MEMIT-LTI demonstrate significant improvements in overfitting metrics (DP and EOS) compared to ROME and MEMIT, indicating effective mitigation of editing overfit.

---

### Official Review · Reviewer_cgEK · 2024-11-03

**Soundness:** 3
**Presentation:** 4
**Contribution:** 3
**Rating:** 8
**Confidence:** 4

**Summary:**

The paper investigates the phenomenon of Editing Overfit within large language models. Editing Overfit occurs when edited LLMs assign disproportionately high probabilities to the edit target, which negatively impacts the model's generalization ability, particularly in complex reasoning tasks. The authors introduce a new benchmark, EVOKE, designed to assess overfitting in LLMs across several challenging scenarios. To address the Editing Overfit problem, the authors propose a strategy, Learn to Inference (LTI), which encourages LLMs to recall edited knowledge in a manner more consistent with their natural inference mechanisms.

**Strengths:**

- The paper clearly identifies Editing Overfit as a pervasive issue within existing LLM editing methods, providing evidence through extensive experimentation and analysis. The identification of this new key problem is one of the major contribution of this paper.
- Besides identifying the problem, the paper also introduces the EVOKE benchmark with fine-grained evaluation metrics, allowing for a more systematic and comprehensive evaluation of the Editing Overfit problem.
-  The paper also carefully analyzes different potential solutions and proposed the Learn to Inference (LTI) approach, an effective plug-and-play module that is highly adaptable for integration with existing knowledge editing methods. The authors also provide a detailed evaluation across multiple LLMs and editing methods, showing that LTI effectively reduces Editing Overfit without compromising editing success.

**Weaknesses:**

- It seems that LTI is only applicable in cases where the edited knowledge can be explicitly represented as knowledge triples (s, r, o, o*). However, more complex or nuanced knowledge editing tasks may not fit into this structured format, potentially limiting LTI's applicability in real-world scenarios.
- The phenomenon of Editing Overfit and the proposed LTI approach are only tested on small-scale LLMs such as GPT-J and GPT-2. Extending the analysis to larger models, such as LLaMA-2/3 or other larger models, would provide a more comprehensive understanding of LTI’s effectiveness at scale.

**Questions:**

- Does knowledge editing on larger models, such as LLaMA-2/3, still suffer from the problem of Editing Overfit? How would the proposed LTI strategy perform on these larger models?
- Are there specific types of facts or domains (e.g., factual knowledge vs. commonsense reasoning) where the problem of Editing Overfit is more severe? Likewise, is there any specific types of knowledge where LTI may be less effective?

---

> ### Author Response · Authors · 2024-11-22
> **Response to Reviewer cgEK**
>
> We highly appreciate your valuable insights and acknowledgment of our contributions. We hope the following comments could address your concerns.
>
> **(W1): It seems that LTI is only applicable in cases where the edited knowledge can be explicitly represented as knowledge triples $(s, r, o, o^{*})$. However, more complex or nuanced knowledge editing tasks may not fit into this structured format, potentially limiting LTI's applicability in real-world scenarios.**
>
> Thank you for raising this insightful point!  We acknowledge that our approach implemented via Multi-stage Constraint, is best suited for edits that can be explicitly represented as knowledge triples (s, r, o, o*), which also the key focus in current research. We agree that enabling models to "Learn To Inference" in more general knowledge contexts is an intriguing topic for future research, and **we've added a dedicated Limitations section in Appendix A of the revised paper to explicitly discuss this limitation.**
>
> **(W2 and Q1): The phenomenon of Editing Overfit and the proposed LTI approach are only tested on small-scale LLMs such as GPT-J and GPT-2. Extending the analysis to larger models, such as LLaMA-2/3 or other larger models, would provide a more comprehensive understanding of LTI’s effectiveness at scale.**
>
> Thank you for your valuable comment! To address the concern, we have extended our experiments to evaluate the performance of baseline methods (FT, FT-L, ROME and MEMIT) and LTI on the **Llama-2-7B** using our EVOKE benchmark (Details results are included in Appendix G), and some results are as follows:
> | Editor          | Efficacy(&uarr;) | Paraphrase(&uarr;) | Prefix-Distraction DP (&darr;) | Prefix-Distraction EOS (&uarr;)| Multihop DP (&darr;) | Multihop EOS (&uarr;) | Subject-Specifity DP (&darr;) | Subject-Specifity EOS (&uarr;)| Relation-Specificity DP (&darr;) | Relation-Specificity EOS (&uarr;) |
> |-----------------|----------|------------|-------------------|-------------------|------------------|----------------|----------------|-----------------|---------------|----------------|
> | Llama-2-7B Base | 13.09    | 15.08      | 16.42             | 58.44             | 1.80             | 86.01          | 0.74           | 97.38           | 1.18          | 82.92          |
> | FT              | 99.61    | 92.48      | 51.48             | 5.61              | 27.71            | 35.40          | 9.74           | 46.07           | 23.12         | 18.83          |
> | FT-L            | 92.73    | 21.53      | 16.90             | 53.17             | 2.59             | 85.16          | 1.12           | 95.20           | 1.54          | 81.08          |
> | ROME            | 100.00   | 93.60      | 22.80             | 46.66             | 21.66            | 63.99          | 21.82          | 45.63           | 2.09          | 81.11          |
> | ROME-LTI        | 100.00   | 89.42      | 17.90             | 53.13             | 15.55            | 70.32          | 14.30          | 57.42           | 1.73          | 81.36          |
> | MEMIT           | 100.00   | 96.80      | 40.76             | 25.36             | 30.47            | 48.18          | 39.09          | 23.36           | 3.87          | 78.08          |
> | MEMIT-LTI       | 100.00   | 91.71      | 26.75             | 36.81             | 14.15            | 67.52          | 9.06           | 47.60           | 2.15          | 80.30          |
>
> From the results, we observe the following conclusions, **which are consistent with our findings on GPT-J and GPT 2-XL models**:
>
> - The baseline methods still demonstrate a pronounced Editing Overfit phenomenon on our EVOKE benchmark. Nearly all successfully edited models exhibit significantly higher Direct Probability (DP) scores compared to the unedited model.
>
> - Both ROME-LTI and MEMIT-LTI demonstrate significant improvements in overfitting metrics (DP and EOS) compared to ROME and MEMIT, indicating effective mitigation of editing overfit.
>
> **(Q2): Are there specific types of facts or domains where the problem of Editing Overfit is more severe? Is there any specific types of knowledge where LTI may be less effective?**
>
> We appreciate the reviewer’s insightful comments! The central point of our paper is that the **Editing Overfit phenomenon likely stems from existing knowledge editing paradigms, which emphasize the direct correspondence between the input prompt $p(s,r)$ and the output $o^*$ for each edit sample $(s,r,o,o^{*})$.** Given that most existing editing methods do not differentiate or focus on knowledge types or domains, and primarily establish input-output mappings, we believe that the Editing Overfit phenomenon likely appears independent of specific knowledge types, while we acknowledge that the extent of overfit could be influenced by knowledge types. We also believe that exploring the performance of various editing models across different types of knowledge domains represents a promising topic for future research.

---

### Official Review · Reviewer_hnCp · 2024-11-04

**Soundness:** 3
**Presentation:** 3
**Contribution:** 3
**Rating:** 8
**Confidence:** 4

**Summary:**

In this paper, the authors focus on investigating parameter-modifying knowledge editing methods and find that existing parameter-modifying knowledge editing approaches exhibit overfitting. Specifically, when questions related to the edited subject and relation arise, the model tends to respond with the edited object, regardless of whether that object is a reasonable answer to the question.

Based on this finding, the authors propose the EVOKE benchmark to assess the degree of overfitting in previous parameter-modifying knowledge editing methods and to evaluate the effectiveness of past techniques designed to mitigate overfitting. They discover that, with the exception of Specificity Augmentation, most methods show limitations.

To address the overfitting issue, the authors introduce a plug-and-play strategy called LTI. Experimental results demonstrate that LTI is highly effective in alleviating overfitting.

**Strengths:**

- The overfitting issue in parameter-modifying knowledge editing methods identified by the authors is an interesting finding.

- The proposed benchmark contributes to a more comprehensive analysis of the effectiveness of knowledge editing methods.

- The authors provide extensive experimental results, conducting a detailed analysis of previous parameter-modifying knowledge editing methods and testing various previously suggested mitigation techniques.

- The LTI method proposed by the authors is a plug-and-play solution, making it convenient to integrate with previous methods for enhanced performance.

**Weaknesses:**

- The proposed LTI method relies on the model's in-context learning capability. This approach requires the unedited model to generate correct answers based solely on the context of the new knowledge, thereby providing an accurate representation and output distribution constraint for the editing process. This implies that for smaller models less proficient in in-context learning, or for models particularly rigid regarding certain facts, the constraints provided by the LTI method may negatively impact editing performance.

**Questions:**

- I am curious whether the representation and output distribution of the unedited model are computed only once. If so, I wonder whether applying Paraphrase Augmentation to this new knowledge would improve performance. The final loss could potentially be adjusted to compute the average KL divergence across different paraphrases. The paper mentions that Paraphrase Augmentation can actually exacerbate overfitting in knowledge editing methods. Therefore, would applying Paraphrase Augmentation to the new knowledge enhance the quality of the constraint, or would it ultimately reduce the effectiveness of the edit?

---

> ### Author Response · Authors · 2024-11-22
> **Response to Reviewer hnCp**
>
> We sincerely appreciate your recognition of our work and thank you for your valuable feedback. We hope the following responses will effectively address your concerns.
>
> **(W1): The proposed LTI method relies on the model's in-context learning capability. This approach requires the unedited model to generate correct answers based solely on the context of the new knowledge, thereby providing an accurate representation and output distribution constraint for the editing process. This implies that for smaller models less proficient in in-context learning, or for models particularly rigid regarding certain facts, the constraints provided by the LTI method may negatively impact editing performance.**
>
> Thank you for your insightful comment regarding the reliance of the proposed LTI method on the model’s in-context learning capability. We agree that this dependence could pose limitations for smaller models less proficient in in-context learning or for models rigid in their handling of certain facts. While acknowledging this dependency, we believe the continued advancements in LLMs and their growing capabilities make approaches like ours, which leverage these inherent strengths, increasingly promising. **We have added a dedicated Limitations section in Appendix A of the revised version.** This section thoroughly discusses these limitations and their implications.
>
> **(Q1) I am curious whether the representation and output distribution of the unedited model are computed only once. If so, I wonder whether applying Paraphrase Augmentation to this new knowledge would improve performance. The paper mentions that Paraphrase Augmentation can actually exacerbate overfitting in knowledge editing methods. Therefore, would applying Paraphrase Augmentation to the new knowledge enhance the quality of the constraint, or would it ultimately reduce the effectiveness of the edit?**
>
> Yes, the representation and output distribution of unedited model are computed only once. **To further answer your question, we apply Paraphrase Augmentation to our LTI strategy (MEMIT-LTI w/ Para).** The results are as follows:
>
> | Editor | Efficacy (&uarr;) | Paraphrase (&uarr;) | Prefix-Distraction DP(&darr;) | Prefix-Distraction EOS(&uarr;) | Multihop DP(&darr;) | Multihop EOS (&uarr;) | Subject-Specifity DP (&darr;) | Subject-Specifity EOS(&uarr;) | Relation-Specifity DP (&darr;)| Relation-Specifity EOS(&uarr;) |
> |---------------------|----------|------------|---------------------|---------------------|-------------|--------------|----------------|-----------------|---------------|----------------|
> | MEMIT               | 100.00   | 95.23      | 18.30               | 39.32               | 25.80       | 50.49        | 33.45          | 17.69           | 0.95          | 82.14          |
> | MEMIT w/ Para       | 100.00   | 98.06      | 22.91               | 35.04               | 39.39       | 35.64        | 44.50          | 7.21            | 1.05          | 82.48          |
> | MEMIT-LTI           | 100.00   | 91.03      | 10.98               | 48.56               | 16.35       | 61.44        | 19.96          | 29.91           | 0.64          | 82.84          |
> | MEMIT-LTI w/ Para   | 100.00   | 94.47      | 12.60               | 46.64               | 31.53       | 41.00        | 34.39          | 12.88           | 0.69          | 82.92          |
>
> From the results, we can observe that **MEMIT-LTI w/ Para** demonstrates a notable reduction in overfitting metrics compared to MEMIT-LTI, indicating that **Paraphrase Augmentation can limit LTI's effective in addressing the Editing Overfit. Nevertheless, the overall overfitting metrics remain superior to those of MEMTI w/ Para, further validating the effectiveness of our LTI in mitigating overfiting.**  We also acknowledge that while Paraphrase Augmentation exacerbates overfitting to some extent, it enhances the correctness of paraphrase-form responses to edited prompts, presenting a trade-off in practical applications. Its usage should therefore be evaluated on a case-by-case basis.

---

> > ### Comment · Reviewer_hnCp · 2024-11-26
> >
> > Thank you for your clarification. I keep the score and increase my confidence.

---

### Public Comment · ~Cai_Yuchen1 · 2024-11-29
**some question**

Dear author:
I read your paper, and it’s well-written, but I have some questions. How do you view the issue of the decline in Paraphrase shown in Figure 7? I believe this is due to the dataset, as the similarity in representation between paraphrase sentences and edited sentences in the dataset is not high. This may be similar to the situation in COUNTERFACT. Generally, I think the current evaluation of paraphrase in the field of knowledge editing is not fair.


Furthermore, I do not consider this to be a flaw in the paper. The authors' explanation of overfitting is very insightful, and their proposed solutions are constructive. As a peer in the field of knowledge editing, I believe this paper deserves to be accepted with a high score.

---

> ### Author Response · Authors · 2024-11-30
> **Thank you for your appreciation and question！**
>
> Thank you for your recognition and interest in our work!
>
> **We agree that the dataset factors you mentioned could indeed be a possible reason for the slight decrease in paraphrase performance observed in Figure 7.** The paraphrase task setting in EVOKE, similar to most existing datasets, follows the COUNTERFACT dataset, where the test questions primarily differ from the original Edit Prompt in terms of sentence structure (expression of relations), while maintaining consistency in the subject form with the original Edit Prompt.
>
> **We also believe that a crucial factor is that the strong paraphrase metrics demonstrated by many baseline models (including ROME and MEMIT) might be partly attributed to overfitting.** For instance, COUNTERFACT's paraphrase tests primarily modify the form of relations, while keeping the subject form unchanged. Since the subject form remains unchanged, models that overfit to the $s
>  &rarr; o^*$ pattern might benefit from this setting: they can easily identify the subject pattern in the question and output $o^{*}$ accordingly. This perfectly satisfies the requirements of the paraphrase task. Therefore, when suppressing model overfitting, we may simultaneously reduce the contribution of overfitting to paraphrase metrics. This could be a significant reason for the results shown in Figure 7, where we observe a slight decrease in paraphrase metrics when applying data augmentation and LTI.
>
> **Furthermore, we agree with your perspective that current paraphrase evaluations might not be comprehensive enough.** Although this may slightly exceed the scope of this paper, some previous works [1] have introduced evaluation tasks that replace the subject $s$ with its aliases, and they found that most editing methods do not perform well in such scenarios. This approach of transforming $s$ contrasts with the relation-focused paraphrasing in most existing works, which we believe indicates that real-world paraphrase tasks might have more complex manifestations, just as how the overfitting studied in this paper might manifest in various complex situations. This could be an interesting topic for future research.
>
> Thank you again for recognizing our work and we welcome further discussions!
>
> [1] Yunzhi Yao, Peng Wang, Bozhong Tian, Siyuan Cheng, Zhoubo Li, Shumin Deng, Huajun Chen, and Ningyu Zhang. 2023. Editing Large Language Models: Problems, Methods, and Opportunities. In Proceedings of the 2023 Conference on Empirical Methods in Natural Language Processing, pages 10222–10240, Singapore. Association for Computational Linguistics.

---

### Meta-Review · Area_Chair_xzkw · 2024-12-20

**Metareview:**

This  paper investigates parameter-modifying knowledge editing methods and identifies that existing approaches exhibit overfitting. Specifically, when questions related to the edited subject and relation arise, the model often responds with the edited object, even if it is not a reasonable answer. To assess overfitting in previous methods and evaluate techniques designed to mitigate it, the authors propose the EVOKE benchmark. Their findings reveal that most methods, except for Specificity Augmentation, have significant limitations. To address this issue, the authors introduce a plug-and-play strategy called LTI, which is shown to be highly effective in alleviating overfitting through experimental results. All reviewers agree that the paper makes a clear contribution. It is recommended that the authors carefully revise the paper according to the reviewers' suggestions and improve the experimental results.

**Additional Comments On Reviewer Discussion:**

All reviewers unanimously agree that the paper can be accepted for publication.

---

### Decision · Program_Chairs · 2025-01-22

Accept (Spotlight)